# Preparation and Properties of Starch–Cellulose Composite Aerogel

**DOI:** 10.3390/polym15214294

**Published:** 2023-11-01

**Authors:** Jihong Huang, Jingyang Gao, Liang Qi, Qunyu Gao, Ling Fan

**Affiliations:** 1Food and Pharmacy College, Xuchang University, Xuchang 461000, China; 2School of Food Science and Engineering, South China University of Technology, Guangzhou 510640, China; gjy312770221@126.com (J.G.); 112201llcctv@scut.edu.cn (L.Q.)

**Keywords:** amylum, aerogel, cellulose, starch, adsorbability

## Abstract

In this study, we conducted research on the preparation of aerogels using cellulose and starch as the primary materials, with the addition of N,N′-methylenebisacrylamide (MBA) as a cross-linking agent. The chemical, morphological and textural characteristics of the aerogels were found to be influenced by the proportions of cellulose, starch, and cross-linking agent that were utilized. An increase in the proportion of cellulose led to stronger adsorption forces within the aerogel structure. The aerogel showed a fine mesh internal structure, but the pores gradually increased with the further increase in cellulose. Notably, when the mass fractions of starch and cellulose were 5 wt% and 1 wt% respectively, the aerogels exhibited the smallest pore size and largest porosity. With an increase in the crosslinking agent, the internal structure of the aerogel first became dense and then loose, and the best internal structure was displayed at the addition of 3 wt%. Through texture analysis and the swelling test, the impact of the proportion of cellulose and MBA on the aerogel structure was significant. Dye adsorption experiments indicated that MBA affected the water absorption and expansion characteristics of the aerogel by improving the pore structure. Lastly, in tests involving the loading of vitamin E, the aerogels exhibited a higher capacity for incorporating vitamin E compared to native starch.

## 1. Introduction

Aerogel is a unique material known for its special properties, including a large number of internal voids, low density, and various potential applications such as dye adsorption, heavy metal adsorption, and drug delivery systems [1]. First developed by Kistler in the 1930s [2], aerogel research has been growing ever since, with a wide range of raw materials being utilized, such as silica oxides, inorganic salts, organic polymers, and nanomaterials [3,4,5,6,7,8]. However, the production of these raw materials often involves the use of concentrated alkaline solutions leading to the generation of wastewater, which poses a significant environmental challenge due to its harmful effects on the natural environment [9,10,11,12,13,14].

Starch is a biomolecule with abundant reserves on earth, offering numerous characteristics that make it highly attractive: readily available raw materials, high biodegradability, low cost, and a simple preparation process. Its molecular structure is characterized by a multitude of reactive hydroxyl groups, particularly the reactive ones that serve as the foundation for starch modification. Leveraging this feature, the hydroxyl groups on the surface of starch aerogels can be utilized as sites for various reactions, thereby expanding the potential applications of starch aerogels [15,16,17]. Given that starch serves as a fundamental component in most food products, it possesses unique advantages as a raw material, enabling it to fulfill essential functions such as drug delivery in vivo and food material encapsulation. Consequently, powder-based aerogels hold significant research value and merit extensive exploration.

Despite its advantages, starch-based aerogel still has some limitations. One of the main drawbacks is its relatively low strength, especially under wet conditions because the exposed hydroxyl groups in the starch molecular chain are easy to combine with water molecules to form hydrogen bonds. When in contact with water, the physical properties of the material are compromised, leading to a decrease in volume [18]. To address these issues and enhance the material’s physical properties, it is necessary to introduce a stable framework within the starch structure.

Cellulose, a polymeric material formed by glycosidic bonds, offers a solution to these challenges. Unlike starch, cellulose possesses a strong molecular composition that is minimally affected by water, maintaining its strength even when immersed in water at temperatures as high as 100 °C [19,20]. Moreover, cellulose shares a similar glucose ring structure with starch, resulting in a high compatibility between the two materials. Through hydrogen bonds and intermolecular forces, cellulose can effectively form composite materials with starch, making it an excellent choice as a support skeleton for starch-based aerogels. At present, there is no report of starch–cellulose composite aerogels, so the effect of the two components on the structural properties of aerogel is worthy of further study

Therefore, this study focuses on the utilization of cellulose and starch, which possess strong biological functionalities, to develop aerogel materials. The objective is to create a composite aerogel by adjusting the ratio of these two raw materials in order to retain their inherent characteristics, namely, ease of preparation, biodegradability, and structural stability. The investigation systematically explores the role of cellulose, starch, and the cross-linking agent methylenebisacrylamide in the formation of aerogels. Additionally, several experiments are conducted to assess various aerogel properties, including morphology, swelling behavior, dye adsorption capacity, and mechanical strength. Lastly, the study explores the potential application of the developed aerogel for vitamin E adsorption.

## 2. Materials and Methods

### 2.1. Experimental Materials and Reagents

Potato starch was obtained from Hua’ou Starch Co., Ltd. (Hohhot, China); Cellulose powder (particle size of 65 μm), N,N′-methylenebisacrylamide (MBA), and methylene blue were purchased from Aladin Biochemical Technology Co., Ltd. (Shanghai, China); and Vitamin E was purchased from Macklin Biochemical Technology Co., Ltd. (Shanghai, China). All other chemical reagents used in this experiment were analytic reagents and purchased from Aladin Biochemical Technology Co., Ltd. (Shanghai, China).

### 2.2. Experimental Methods

#### 2.2.1. Preparation of Composite Aerogel

Firstly, the water content of potato starch (PS) was measured by an IR-35 infrared rapid moisture analyzer (Denver Co., Ltd., Denver, CO, USA). Then, a specific amount of PS and microcrystalline cellulose was taken in a beaker with 100 mL deionized water, resulting in a combined mass fraction of 6 wt% (dry basis) for the two materials. Then, a certain amount of the crosslinking agent MBA was added to the above solution (the mass fraction was 0 wt%, 0.5 wt%, 1 wt%, 3 wt%, 5 wt% of dry basis) followed by the composite initiator (NH_4_)_2_S_2_O_8_ and NaHSO_3_ (the added weight was 1% and 2% of MBA, respectively).

In the crosslinking process, the starch particles were gelatinized in the aqueous solution to fully release molecular chains and expose active hydroxyl groups. Then, the initiators were added to the starch paste to initiate the free radical compolymerzation. They were decomposed to generate initiating radicals, such as SO_4_^−^, OH, and HSO_3_^−^, which attracted the hydrogen atoms attached to the accessible starch hydroxyls and created starch macroradicals. Cross-linker MBA with unstable double bonds could graft onto the starch molecule chains by covalent bonding and further form long chain polymers by copolymerization. This way, cross-linking can take place under relatively mild conditions (room temperature and pH 7.0) to form a network structure. See the Appendix A for more details on the cross-linking process (Appendix A).

The solution was subsequently placed in a water bath at 90 °C for 0.5 h while being magnetically stirred at a rate of 1.67 r/s. Once the treatment was complete, the mixture was directly transferred into a 20 × 20 × 15 mm^3^ container, and the evaporation of water was prevented by covering it with cling film. After allowing the mixture to cool to room temperature, it was placed in a 4 °C environment for 24 h to form a hydrogel. Subsequently, the material was placed in a −18 °C environment for another 24 h. Finally, the frozen gel was removed from the mold and subjected to a 24-h freeze-drying process. The resulting composite aerogel was labeled based on the proportion of its components, denoted as S(starch)*C(cellulose) or S*C*B (cross-linker MBA). For example, S5C1B3 indicates that the composite aerogel consists of 5 wt% starch, 1 wt% cellulose, and 3 wt% cross-linker.

#### 2.2.2. Morphological Properties of Composite Aerogels

The material was positioned next to a measuring tape and captured by camera to acquire an image depicting its external structure. The composite aerogel was segmented into multiple thin layers, approximately 0.1 cm thick, enabling microscopic examination using a Merlin scanning electron microscope (Zeiss Co., Oberkochen, Germany) at an accelerating voltage of 5 kV. The magnification was 100×. One of the layers was carefully positioned on the microscope stage, securing its upper section to the carrier table using a conductive adhesive, followed by a 5-min gold coating using an MCM-100 Ion Sputter Coater (Jing Teng Tech Co., Ltd., Beijing, China). Subsequently, the material and the carrier table were positioned within the electron microscope chamber, where the air was evacuated prior to observation.

The average droplet size and size distribution was determined through statistical analysis from optical microscope images using Image-pro Plus 6.0.

#### 2.2.3. Analysis of the Functional Groups of Composite Aerogels

The aerogel was ground into a powder, 3 mg composite aerogel powder and 200 mg potassium bromide powder were mixed well in a drying oven, and the dried mixture was pressed into tablets under a pressure of 20 Mpa at room temperature for 30 s. The obtained lamellar material was analyzed under an infrared spectrometer (Thermo Fisher Scientific Co., Walthman, MA, USA) between 4000 and 400 cm^−1^ with 64 scans at a spectral resolution of 4 cm^−1^.

The NMR spectra were recorded using an AVANCE digital 400 spectrometer (Bruker, Frankfurt, Germany) operating at 400 MHz for 1 H NMR spectroscopy and 100 MHz for 13C NMR spectroscopy, respectively. The 1 H spectra were recorded in 128 individual scans with a sweep width of 16 ppm and a delay time of 1 s. Prior to NMR analysis, 5 mg composite aerogel powders were dissolved in deuterium dimethyl sulfoxide (DMSO-d6) at 40 °C to obtain clear solutions.

#### 2.2.4. Texture Analysis of Composite Aerogels

The determination of textural characteristics was conducted using a TA-XT2 type physical property tester (Stable Micro Systems Co., Ltd., London, UK). The prepared starch aerogel cubes were used as the samples, with one side measuring 20 × 20 mm^2^ and placed facing downwards. TPA (Texture Profile Analysis) testing was performed using a P/36R probe. The pre-test speed, test speed, and post-test speed were set at 3.0 mm/s, 1.0 mm/s, and 3.0 mm/s, respectively. The compression ratio was set at 50%, and the trigger force was 5 g. The compression was performed at intervals of 5 s, and the test was repeated at least 4 times under the same conditions. The data were fitted using the MACRO program in the Exponent system to obtain the hardness, viscosity, and elasticity indices of the samples.

#### 2.2.5. Swelling Properties of Composite Aerogels

First, the mass of the composite aerogel was determined using an electronic balance. The prepared sample was placed in a beaker containing deionized water at room temperature (25 °C). After soaking for 0.5 h, the total mass of the material was measured, and the swelling ratio was obtained by Equation (1). The unit is expressed as g/g. This process was repeated at 50, 70 and 90 °C. Finally, the swelling ratio of the composite aerogel at four different temperatures was summarized and recorded.
(1)$$\mathrm{Swelling}\;\mathrm{ratio}=\frac{M_{t}}{M_{0}}$$
where *M*_0_ (g) represents the mass of the initial unsoaked sample; and M*_t_* (g) is the total mass of sample after soaking in water at corresponding temperature.

#### 2.2.6. Dye Adsorption of Composite Aerogels

Firstly, 50 mL of an aqueous solution containing methylene blue at a concentration of 10 g/L was added to a 100 mL centrifuge tube along with a magnetic stirrer. Then, a specific mass of cross-linked starch aerogel was accurately weighed. The aerogel was introduced into the centrifuge tube, which was subsequently placed on the magnetic stirrer inside a water bath maintained at 25 °C to initiate the experiment. Once the experiment commenced, aliquots of the solution were extracted at the specific time intervals of 0 h, 1 h, 2 h, 4 h, 8 h, 12 h, and 24 h. Each of the seven extracts was then diluted to one-tenth of the original concentration, and the corresponding absorbance was measured and recorded. Based on the obtained data, absorbance curves for the extracts were plotted. Similarly, a standard curve for methylene blue was constructed using solutions of 0.01 g/L, 0.005 g/L, 0.0025 g/L, 0.001 g/L, and 0.0005 g/L. The instrument wavelength was adjusted to 660 nm, and a cuvette filled with deionized water served as the reference solution in the first compartment of the cuvette holder. The measurement of the adsorption of methylene blue refers to the amount of methylene blue material adsorbed per unit weight of the aerogel material. The unit is expressed as mg/g.

#### 2.2.7. Analysis of Fat-Soluble Vitamin Adsorption of Composite Aerogels

Five composite aerogel samples were chosen to assess the adsorption capacity of fat-soluble Vitamin E. A standard curve for Vitamin E was constructed using solution with concentrations of 0.05, 0.1, 0.2, 0.4 and 0.5 mg/mL. Initially, 25 mg Vitamin E was dissolved in 50 mL anhydrous ethanol to achieve the concentration of 0.5 mg/mL. 0.1 g of the samples were then immersed in the prepared solution and shaken for 30 min to ensure the complete adsorption of Vitamin E. The adsorption capacity of vitamin E in various types of samples was calculated based on Equation (2).
(2)$$\mathrm{Adsorption}\;\mathrm{capacity}=\frac{\left(C_{0}-C_{t}\right)V}{M}$$
where C_0_ and C_t_ (mg/mL) represents the concentration of Vitamin E before and after the adsorption process; V (mL) is the volume of the solution; and M (g) is the mass of sample.

## 3. Experimental Results and Discussion

### 3.1. Morphological Properties of Composite Aerogel

Figure 1 displays the SEM images of the various composite aerogel samples, magnified 100× under electron microscopy. The relationship between the starch and cellulose content in the samples could be observed based on their respective designations in the experiment: S6C0, S5C1, S4C2, and S3C3. Analysis of the SEM images revealed that changes in starch content impacted the pore diameter of the aerogel, with lower starch content resulting in larger pores. In fact, the pore diameter of S6C0 was even smaller than that of S3C3. Furthermore, the SEM images indicated that the aerogels in S5C1 and S4C2 exhibited denser pores. Despite having smaller diameters, a larger unit surface area was achieved with a lesser amount of starch. All pore size data are provided in the Appendix A.

Cellulose, being insoluble in water, primarily served a supportive and stabilizing role within the aerogel structure. When starch was added, the rigid cellulose molecule chains built more connection points between the free starch molecules (within the gelatinized starch), forming a denser internal structure. The SEM images of S4C2 and S3C3 revealed residual cellulose (irregular lumps as indicated by the arrows) within the aerogel pores. In this case, the cellulose primarily strengthened the starch structure and did not act as a connection site. The reason for this occurrence was that as starch content decreases, some cellulose remains unwrapped by starch molecules, thereby exacerbating pore enlargement. Additionally, increased cellulose content hampers the formation of separate pores. Consequently, we concluded that the starch-to-cellulose ratio in S5C1 is the most suitable. It could be seen that S5C1 was a more appropriate ratio, and the aerogels formed at this ratio were the densest and the cellulose was highly utilized.

Figure 2 depicts the scanning electron images of aerogels produced by incorporating varying amounts of N,N′-methylenebisacrylamide (MBA) as a cross-linking agent. A comparison of the SEM images revealed the direct impact of MBA on the internal structure of the aerogels. Notably, when the percentage of MBA was 0.5% and 1%, the pore diameters of the resulting aerogels (Figure 2a,b) were significantly larger than those in S5C1B3 (Figure 2c). These findings indicated that the pore size gradually decreased with increasing MBA content up to 3%. This phenomenon could be attributed to MBA that builds more connection points between the free starch molecules in the paste, forming a denser internal structure.

When the MBA proportion reached 5%, the aerogel began to exhibit laxity, resulting in an increase in pore diameter. On one hand, the addition of a small quantity of hydrophilic MBA led to more water content, causing the formation of larger ice crystals during the freezing process [21,22]. On the other hand, excessive cross-linking adversely affected the hydrogen bonding force between molecules, reducing molecular spacing and resulting in the looser internal structure of the composite aerogel [23]. Similar conclusions were drawn from other experiments, highlighting the trend of the pore structure becoming denser initially and then looser as the proportion of cross-linking agent MBA progressively rose [24]. Therefore, achieving the desired aerogel structure necessitates maintaining an appropriate ratio of starch to cellulose during hydrogel formation.

### 3.2. The Fourier-Transform Infrared Spectroscopy (FTIR) of Aerogels

Figure 3a illustrates the IR spectra of the S5C1B1, S5C1, S6C0, and S0C6 aerogels. It could be observed that all of the aerogels exhibit a broad peak at 3320 cm^−1^, which corresponded to the characteristic absorption peak of starch and cellulose (the stretching vibration of -OH groups). The stretching vibration generated by the saturated hydrocarbon C-H in glucose gave rise to a distinct sharp peak at 2900 cm^−1^, which was also a characteristic absorption peak of starch and cellulose.

In the absence of the cross-linker MBA, no significant change in the fingerprint region was found in the IR spectra of S5C1, S6C0, and S0C6. There were subtle differences between the peak intensities and widths in the spectrum, which could be explained by differences in the hydrogen bond formation and water content between the components. In contrast, S5C1B1 containing the cross-linker displayed a new small peak at 1547 cm^−1^ that was related to the bending vibrations of the N-H bonds on the MBA chains [25]. This phenomenon provided additional evidence that the cross-linker MBA had indeed interacted with starch and cellulose, thereby influencing the structure of the aerogel.

The ^1^H NMR spectra of the samples are shown in Figure 3b. Peaks arising from the anhydroglucose in the starch and DMAEMA were assigned. H-1,2,3,4,5,6 protons for anhydroglucose, along with the H-1′,2′,3′ protons for MBA, were observed. It showed that a slight decrease (from 3.92 to 3.76, calculated by our previous reports [26,27]) was found in the area of the protons for H-2,3,4,5 from S5C1 to S5C1B1, which meant that part of protons of H-2,3,4,5 were replaced by functional monomers. Meanwhile, the area of the protons for H-6 was slightly reduced (from 0.99 to 0.92). This indicated that all the reactive hydroxyls in anhydroglucose were involved in the copolymerization, which was in line with the mechanism analysis (Appendix A).

### 3.3. Textural Analysis of Aerogels

In this study, the hardness, elasticity, and cohesion of the aerogels formed at different starch/cellulose ratios were systematically analyzed using a mass spectrometer (TPA), and the data are detailed in Figure 4. Based on the data, the hardness values, in ascending order, were as follows: 3144 N/m^2^ (S3C3) < 3909 N/m^2^ (S4C2) < 6768 N/m^2^ (S5C1) < 9345 N/m^2^ (S6C0). Therefore, it could be inferred that the hardness of aerogels could vary significantly depending on the amount of starch present in them. It was observed that cellulose and starch served as the main precursor substances in aerogels. During the aging and pasting of the starch, the molecular structure changed from a dispersed state to a reconnected state, which played a crucial role in the overall network structure. Notably, cellulose was distributed discretely within the aerogel and lacked a continuous three-dimensional structure. As a result, it was challenging for cellulose to effectively support the aerogel, despite its more stable internal structure. This limitation also hindered further improvement in the hardness of the aerogel.

Likewise, S3C3 exhibited the lowest cohesion (0.18) and elasticity (0.22) due to its lower starch content. Starch played a vital role in providing support throughout the entire aerogel structure, and a higher starch content significantly affected the aerogel’s structure. Cohesion could be defined as the internal binding strength. The analysis revealed that the elasticity and cohesion of the samples remained stable (cohesion of 0.22–0.23 and elasticity of 0.27), except for S3C3, which experienced a significant decrease. This indicated that elasticity and cohesion were intrinsic properties of starch and exhibit less variation within certain ranges of starch composition. The decrease in elasticity and cohesion after reaching a certain threshold value was influenced by the decrease in starch concentration or the increased proportion of cellulose that hampered the rearrangement process of the starch molecules.

The TPA data obtained at various MBA contents are presented in detail in Figure 5. There was a positive correlation between the MBA content and the hardness value of the aerogel. The data indicated 6173 N/m^2^ (S5C1B0.5) < 6376 N/m^2^ (S5C1B1) < 8842 N/m^2^ (S5C1B3) < 9531 N/m^2^ (S5C1B5). Thus, the MBA content played a crucial role in the formation of aerogels, and its addition contributed to the enhancement of their structural properties.

Regarding cohesion and elasticity, most samples (excluding S5C1B3) exhibited a noticeable decreasing trend with an increasing amount of MBA. In the absence of MBA (data in Figure 4), cohesion and elasticity did not undergo significant changes, despite a significant decrease in hardness. Unlike hardness, cohesion and elasticity were mainly affected by the length and arrangement of starch molecular chains and interaction forces and varied little with MBA amount within a certain range. However, the excessive introduction of the cross-linking agent enhanced the cross-linking effect between the starch molecules. As a result, the aerogel became less prone to easy recovery when compressed, leading to poor cohesion and elasticity (decreased by 14% and 28%).

Moreover, it was observed that the S5C1B3 sample displayed a distinct cohesion and elasticity profile. This could be attributed to the reduced pore size of the aerogel. The smaller pore size in the S5C1B3 sample resulted in a decreased distribution density of the cross-linker, which in turn reduced the structural constraints on the aerogel, leading to improved cohesion and elasticity.

### 3.4. Swelling Properties of Composite Aerogels at Different Temperatures

The swelling properties of different composite aerogels are detailed in Figure 6. It was observed that the swelling ratio of the starch–cellulose composite aerogel samples surpassed that of S6C0 at temperatures below the pasting temperature. However, it was lower than S6C0 when the temperature exceeded the pasting temperature. The incorporation of cellulose in the composite aerogel partially replaced starch, resulting in variations in the internal structure depending on the cellulose content. The water absorption of starch was restricted by the hydrogen bond between the starch and water molecules below the pasting temperature, thereby influencing the swelling ratio of the aerogel to some extent. Additionally, due to the relatively small pore diameter of the aerogel, the presence of cellulose enhanced the swelling ratio compared to S6C0. As the temperature surpassed the pasting temperature (e.g., 70 °C), the swelling ratio of the starch molecules in the pasted state gradually intensified. The internal structure of cellulose remained stable in hot water, therefore, the starch content directly impacted the volume of the aerogel above the pasting temperature. The data indicated a direct relationship between the starch content and the swelling ratio at a temperature of 70 °C. It was important to note that upon a continuous temperature increase to 90 °C, some fragmentation occurred among the samples with lower starch content, leading to the exclusion of the swelling ratio data at that temperature. Consequently, the uncross-linked aerogel samples exhibited low internal structural stability at high temperatures.

Figure 7 illustrates the swelling ratio trend of the produced aerogels at four temperatures (25 °C, 50 °C, 70 °C, and 90 °C) with varying cross-linker contents. The analysis of the data indicated that the water retention capacity of the formed aerogel samples decreased as the cross-linking agent content increased. Consequently, it could be deduced that the incorporation of the cross-linking agent MBA partially hampered the swelling ratio of the composite aerogel, as the cross-linking reaction affected the strength of intermolecular hydrogen bonding.

### 3.5. Dye Adsorption Properties of Composite Aerogels

The curves depicting the adsorption of methylene blue over time for different starch-to-cellulose ratios are illustrated in Figure 8. It was evident that the initial starch aerogel exhibited the poorest adsorption performance. However, after the substitution of cellulose, the adsorption capacity of the sample improved, and there was a consistent increase in adsorption performance with higher cellulose content. Despite the fact that the introduction of cellulose powder relaxed the internal structure of the aerogel, it concurrently provided a larger specific surface area. This was attributed to the alteration of pore diameter, which enlarged as more cellulose was added. Additionally, the cellulose powder maintained its inherent morphology even after starch pasting, leading to an increased specific surface area for the composite aerogel as the proportion of cellulose powder rose. Notably, the S3C3 sample exhibited the highest adsorption capacity primarily due to the larger specific surface area of the cellulose powder. Although the addition of cellulose increased the size of the pores, it also enhanced the adsorption capacity of the aerogel.

The relationship between methylene blue adsorption and time at different cross-linker contents is depicted in Figure 9. Comparing the adsorption of the S5C1 and S6C0 samples, it could be observed that the adsorption capacity of methylene blue gradually decreased as the amount of cross-linker increased. This could be primarily attributed to the significant structural changes induced by the addition of the cross-linking agent. On one hand, the addition of MBA increased the porosity of the aerogel and decreased the pore size, which was conducive to the adsorption of guest molecules. On the other, Figure 7 indicates that the water retention capacity of the formed composite aerogel samples was higher than that of raw aerogel at room temperature, which meant that more methylene blue remained in the aerogel and could not be released quickly. Consequently, S5C1B0.5 exhibited a higher adsorption capacity. As the amount of cross-linker was progressively increased, it exerted an inhibitory effect on the water absorption and swelling of starch due to the formation of strong chemical bonds. The water absorption and swelling of starch not only occurred at the molecular level, but also involved the entire external structure. With the increasing amount of cross-linker, the swelling was constrained, resulting in a reduction in adsorption. Ultimately, this led to a discrepancy between the specific surface area and the adsorption capacity.

### 3.6. Fat-Soluble Vitamin Adsorption Capacity Test

Figure 10 illustrates the results of the fat-soluble vitamin adsorption capacity test conducted on five different composite aerogel samples. The analysis indicated that starch exhibited the lowest loading of vitamin E, with an adsorption of 0.0365 mg/g, significantly lower than the other four composite aerogels. This result suggested that aerogels with a larger specific surface area had a higher capacity for adsorbing vitamin E. Furthermore, S5B1 demonstrated the highest adsorption capability for vitamin E, reaching 0.2355 mg/g at a solution concentration of 50 mg/mL. Among the cross-linked starch aerogel samples, the lowest adsorption was observed in the S5C1B1 sample, which only achieved 0.2237 mg/g. Therefore, it was evident that cross-linked starch aerogels had a greater ability to adsorb vitamin E compared to the original starch granules. Compared with water, anhydrous ethanol had a weak ability to penetrate the molecular chain of starch and could not effectively swell the aerogel. Therefore, there was no significant difference in the volume change of aerogel during the adsorption process, which was why there was no significant difference in the adsorption amount of vitamin E among these samples.

## 4. Conclusions

In this study, cellulose and starch were utilized as raw materials to investigate the properties and correlation of aerogels formed under the influence of varying doses of MBA. FTIR and ^1^H NMR provided evidence that the cross-linker MBA had indeed interacted with starch and cellulose. The aerogel showed a fine mesh internal structure, but the pores gradually increased in size with further increases in cellulose. The aerogels formed with a 3% addition of MBA exhibited the densest pore structure, consequently providing the largest porosity. Excessive amounts of the cross-linking agent hindered the swelling property of the aerogel to some extent, resulting in an initial increase and subsequent decrease in the adsorption capacity of the aerogel as the dosage of the cross-linking agent increased. The aerogel S5C1B0.5 showed good dye adsorbability, but too much of the cross-linking agent would inhibit its ability. Finally, based on experiments involving vitamin E adsorption, it was concluded that cross-linked starch aerogel S5B1 exhibited a significantly higher ability to adsorb vitamin E compared to S6C0. It can be predicted that starch–cellulose aerogel will have important application value in special fields such as drug delivery and food material loading in vivo.

## Figures and Tables

**Figure 1 polymers-15-04294-f001:**
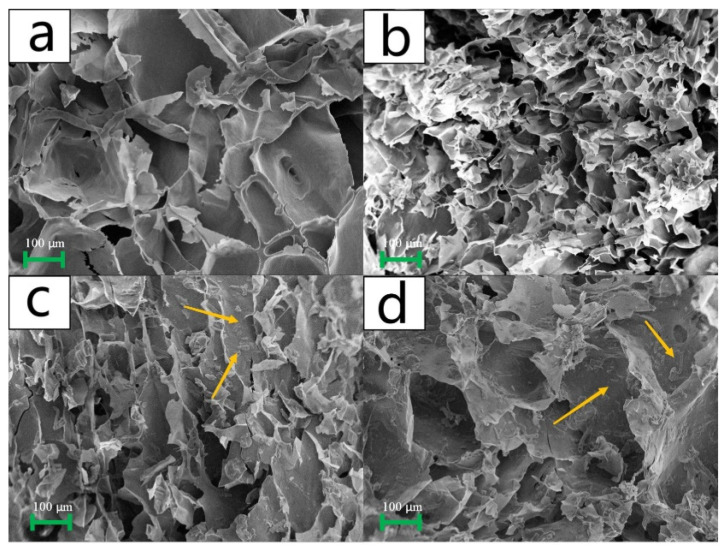
Scanned images (SEM images) of different composite aerogels magnified 100 times, where figures (**a**–**d**) represent the SEM image of S6C0, S5C1 aerogel, S4C2 aerogel, and S3C3 aerogel, respectively.

**Figure 2 polymers-15-04294-f002:**
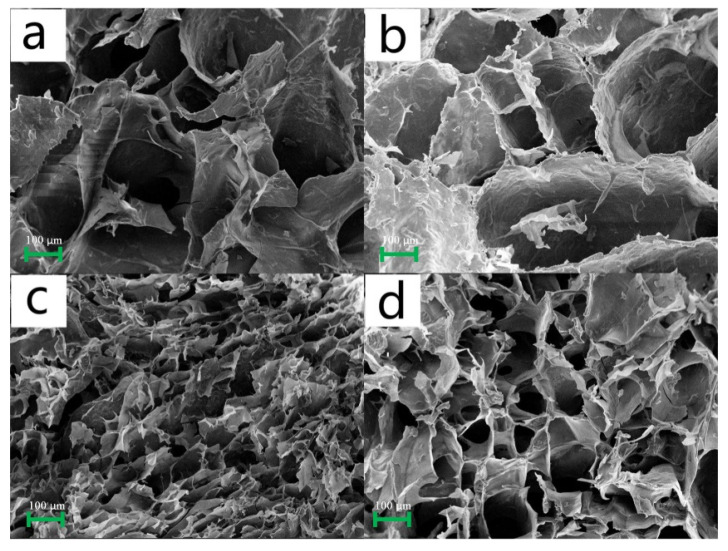
Scanned images (SEM images) of different composite aerogels at 100 times magnification, where figures (**a**–**d**) show the SEM images of the S5C1B0.5 aerogel, S5C1B1 aerogel, S5C1B3 aerogel, and S5C1B5 aerogel, respectively.

**Figure 3 polymers-15-04294-f003:**
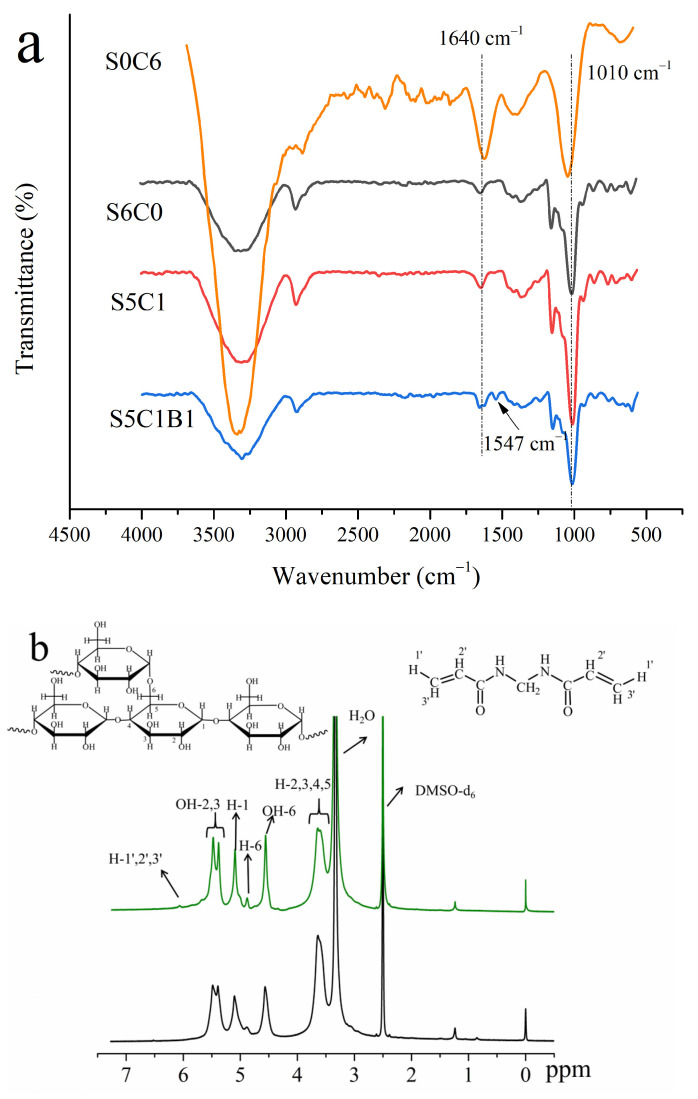
(**a**) Infrared spectra of S5C1B1, S5C1, S6C0 and S0C6 aerogels. (**b**) ^1^NMR of S5C1B1 and S5C1.

**Figure 4 polymers-15-04294-f004:**
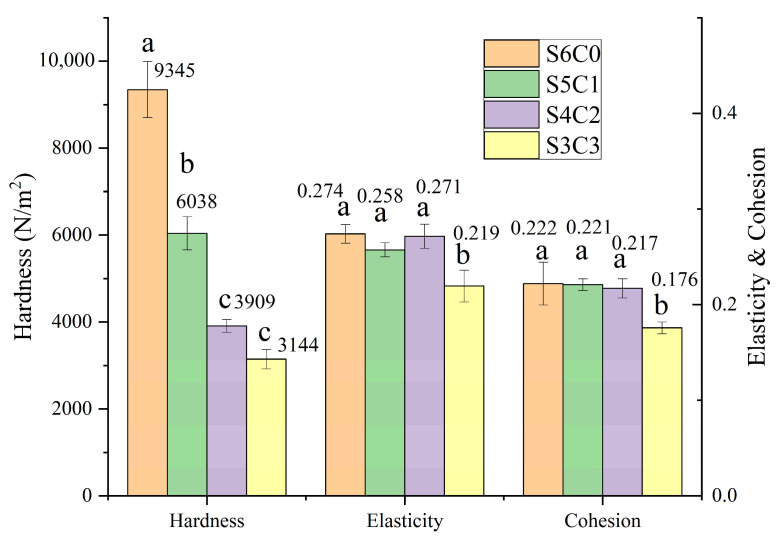
Physical characteristics of aerogels with different starch/cellulose ratios, including: hardness, elasticity, and cohesion. Different letters indicate significant differences of hardness, elasticity and cohesion among the various samples.

**Figure 5 polymers-15-04294-f005:**
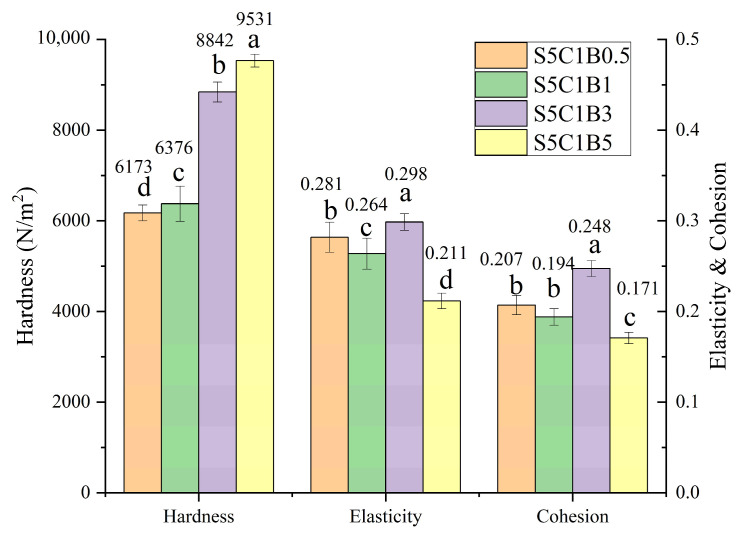
Physical characteristics of aerogel samples at various crosslinker MBA contents. Different letters indicate significant differences of hardness, elasticity and cohesion among the various samples.

**Figure 6 polymers-15-04294-f006:**
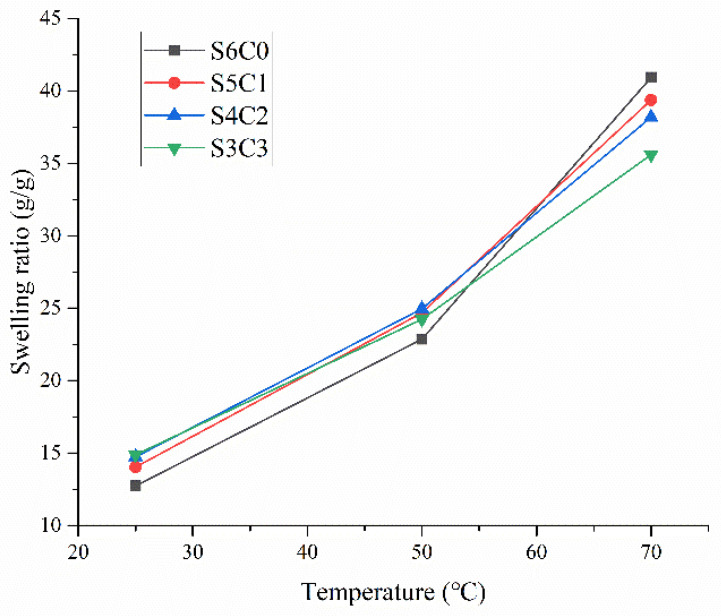
Swelling ratio of different composite aerogels at different starch/cellulose ratios.

**Figure 7 polymers-15-04294-f007:**
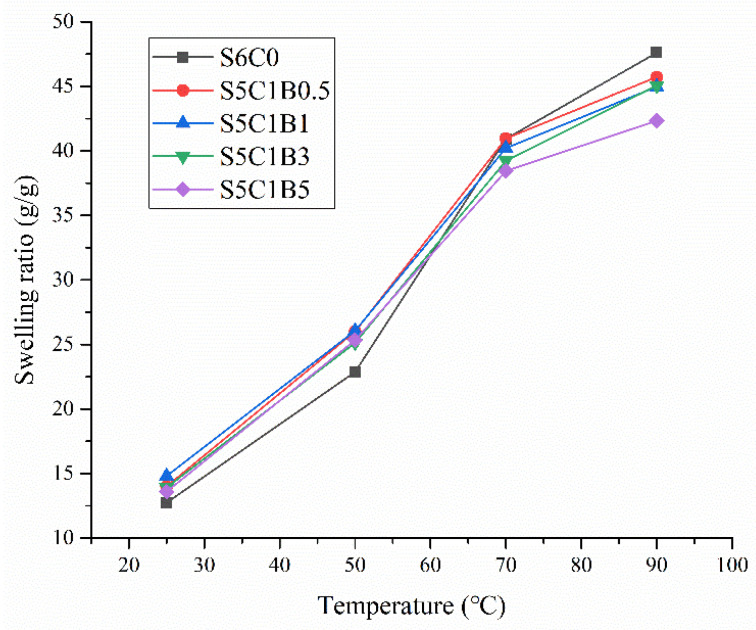
Swelling ratio of the composite aerogel at varying cross-linker contents.

**Figure 8 polymers-15-04294-f008:**
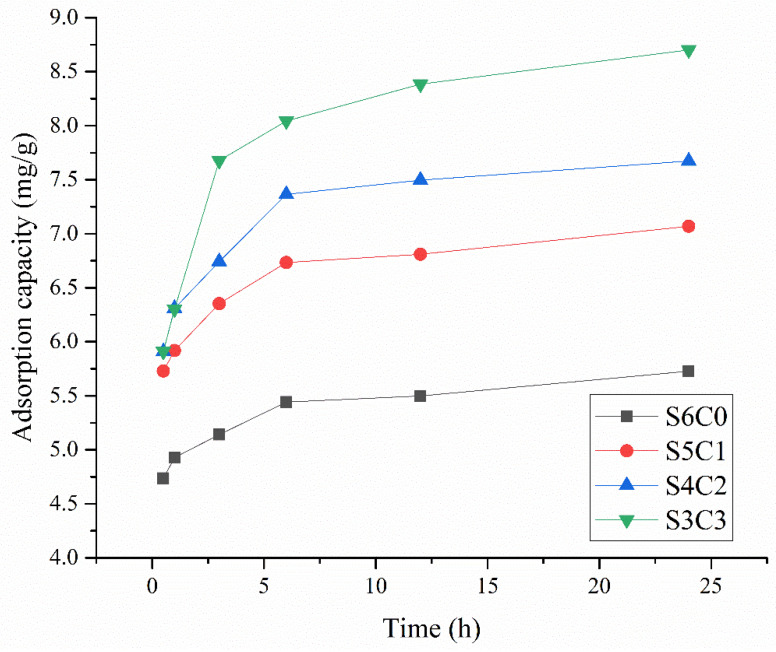
Adsorption of methylene blue by different composite aerogels.

**Figure 9 polymers-15-04294-f009:**
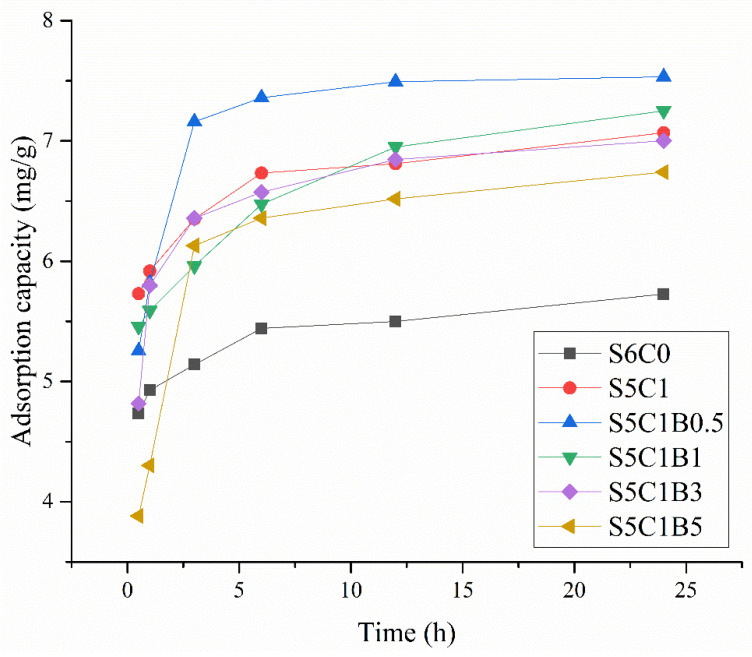
Adsorption of methylene blue in the composite aerogel formed by adding different doses of MBA.

**Figure 10 polymers-15-04294-f010:**
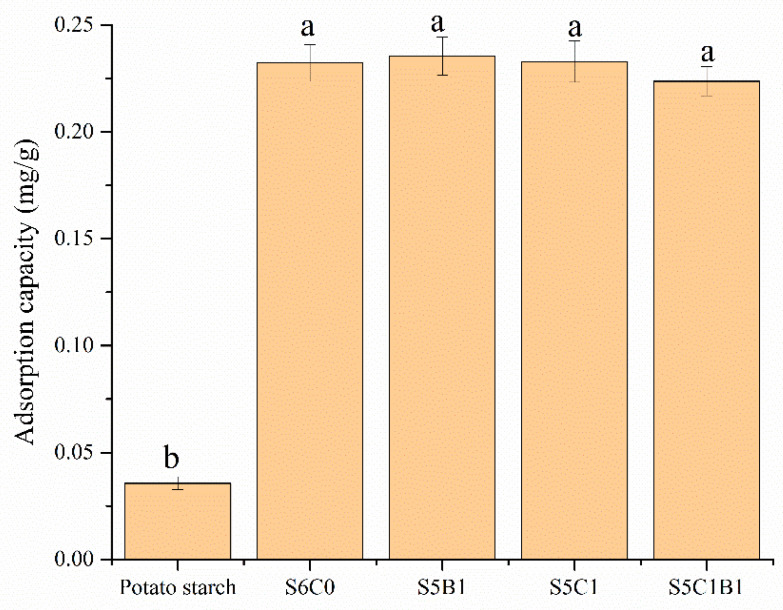
Adsorption of fat-soluble vitamin E by different composite aerogels. Different letters indicate significant differences of adsorption capacity among the various samples.

## Data Availability

The data that support the findings of this study are available from the corresponding author, Qunyu Gao and Jihong Huang, upon reasonable request.

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
