# Peer review of "Preparation and Properties of Starch–Cellulose Composite Aerogel"

_polymers, 2023, doi:10.3390/polym15214294_

Round 1
Reviewer 1 Report (New Reviewer)
Comments and Suggestions for Authors
The manuscript considering the processing of the starch-cellulose hydrogels from Huang et.al scientifically fair. It discusses and points out important and concepts of utilizing cost-effective, nature-derived precursor for the synthesis of useful thermosets with a solid hypothesis to expand the scope of bio-based material development. However, certain concepts, material preparation techniques, and testing techniques are not clearly defined through the manuscript. I endorse it for publication in the Polymer Journal of MDPI upon addressing following comments property.

Manuscript is written via clear and understandable English. However, it is very obvious that the manuscript is reviewed/edited by the PI (red highlights). The red highlights are written with more coherent English which makes it sound like it is written by two different people.
Author Response
Please see the attachment

Reviewer 2 Report (New Reviewer)
Comments and Suggestions for Authors
This manuscript described the study of cellulose and starch as raw materials of varying ratio and conducted various characterizations. Also the authors introduced N,N’-methylenebisacrylamide as a cross-linking agent at varying dose and discussed the properties. However, the manuscript raises many questions and concerns which may potentially negate the validity and value of the works. Meanwhile, the manuscript lacks novelty and depth in mechanism discussion. Therefore, I think this manuscript is not suitable for the publication at Polymers.
Specific comments are shown as follows:
Both experiment process and raw materials are commonly used which lack novelty.
1. The authors mentioned one of the reasons they choose cellulose as raw material is because: “Unlike starch, cellulose possesses a strong molecular composition that is minimally affected by water, maintaining its strength even when immersed in water at temperatures as high as 100°C”. However, the cellulose chain contains abundant hydroxyl groups and could interact well with water. The water absorption will lead to a reduction in stiffness of cellulosic fibres.
2. The authors continuously mentioned that the addition of cellulose and MBA increase the specific surface area and have the influence of the pore size, but there are no characterization/results to support these statements. The authors need to run BET to support their conclusion.
3. There is no sufficient discussion about cross-linking mechanisms, why choose the MBA as cross-linking agent, how does the MBA react with cellulose and starch.
4. The authors should always provide scale bar in SEM pictures.
5. In Figure 3, the -OH peak intensity if S0C6 is much higher than other samples, it cannot be concluded as “no significant change”.
6. The depth discussion of their observation/results are needed.
Author Response
Please see the attachment

Reviewer 3 Report (New Reviewer)
Comments and Suggestions for Authors
Please see the attached file for my comments

The language and parts of the text can be marginally improved.
Round 2
Reviewer 2 Report (New Reviewer)
Comments and Suggestions for Authors
The authors had carefully addressed the concerns and suggestions raised during the initial review process. The revisions have significantly improved the quality of the manuscript, only several minor revisions need to be take care:
1. For the pore size discussion, ImageJ software can be used to measure the pore size dimensions in the SEM picture. Pore size data can express the conclusions more intuitively.
2. Current scale bars in SEM pictures are hard to read. The authors can make the them bolder.
3. Please add the cross-linking mechanism schematic into the text. This can be combined with the FTIR section and made readers easy to follow.
Author Response
Please see the attachment.

This manuscript is a resubmission of an earlier submission. The following is a list of the peer review reports and author responses from that submission.
Round 1
Reviewer 1 Report
Comments and Suggestions for Authors
The manuscript investigated the properties of aerogels obtained from varying levels of starch, cellulose and cross-linker. The topic is of great importance and interesting results are reported. However, the manuscript has some problems and is not acceptable in its present form:
Comments:
Line 99: Please add the model of SEM, magnification, voltage, sputter coater model, and other details.
Line 113: Please explain which textural test was performed (Based on the reported results you must have performed Texture Profile Analysis (TPA)). Also explain which textural parameters were determined and how these parameters are calculated. You can use the following manuscript: 10.1016/j.carbpol.2020.116406
Line 114: Please delete “The material is analyzed by a mass spectrometer and a physical property analyzer”. It is not about texture measurement.
Figure 3: Please ad the FTIR spectra of starch and cellulose
Figures 4 and 5: The hardness of gels is very high but the values reported for cohesiveness and elasticity are very low! It shows that the gels were so fragile or the compression (%) was very high and was not suitable for this test. Please explain why these parameters are very low or change the conditions and perform the tests.
Figures 4 and 5: Add the unit for hardness.
Conclusion: Please explain the potential applications of these aerogels in the conclusion
Comments on the Quality of English LanguageThe language should be edited by a Professional English editor.
Reviewer 2 Report
Comments and Suggestions for Authors
The manuscript polymers-2513707 presents the preparation and characterization of aerogels based on cellulose and starch, by using N,N'-methylenebisacrylamide as a cross-linking agent.
The manuscript still needs a lot of improvements, and some of the problems I noticed are listed below:
- There is a problem with writing the references in the text! These must not appear as a superscript! Please review the references throughout the manuscript!
Keywords
- If the authors used the term "starch" in the manuscript, it would be good for the same term to appear in Keywords. Please review the keyword!
Introduction
- There is no information related to cellulose-based aerogels, even if there is consistent data in literature! Please add in Introduction section also data related this subject, emphasizing the characteristics of this materials! The authors cannot discuss only the aerogels obtained from starch, omitting the information (advantages and disadvantages) regarding cellulose-based aerogels!
2. Materials and Methods
2.1. Experimental materials and reagents
- L. 80: Please add more information related to cellulose, including company and country! The same for vitamin E!
2.2. Experimental methods
2.2.1. Preparation of composite aerogel
- L. 96-97: “S*C* (material without B)”? What is B? Please explain all the abbreviations within the manuscript, at their first use! A table listing the quantities and component materials of the aerogels would be helpful to understand notations used by the authors!
2.2.2. Morphological properties of the composite aerogel
- L. 106: Please add information about the name of SEM microscope, as well as company and country!
2.2.3. Analysis of functional groups of composite aerogels
- The authors must refer to FTIR spectroscopy, but nothing related to this is presented in description! Please revise carefully all the methods, highlighting the equipment and the methodology used to obtain the information!
2.2.4. Texture analysis of composite aerogels
- This method is written ambiguously! Please rewrite it completely and mention the equipment used!
2.2.5. Solubilization properties of composite aerogels
- L. 120-130: “Solubilization properties” or Swelling properties? Please make the adequate correction!
- Please use the Equation Editor to write all the equations used in this manuscript!
2.2.7. Analysis of fat-soluble vitamin adsorption of composite aerogels
-L. 148-162: To determine the “The determination of vitamin E loading” the authors must measure the remained solution after immersion of the aerogels. What the authors established was the release of Vitamin E from the aerogels!
3. Experimental results and discussion
3.1. Morphological properties of composite aerogel
- L. 168: “NPA (S6C0)”? There are two notations for the same sample! Please use only S6C0, in order not to mislead the readers!
- L. 178: “revealed residual cellulose within the aerogel pores”? Please indicate on Figure 1 with an arrow where the authors identify cellulose!
- L. 187-189: “the a figure shows the SEM image” must be “(a) SEM image”!
- L. 215-217: Please make the same corrections for Figure 2!
- L. 202-203: “The addition of a small quantity of hydrophilic MBA during the freezing process led to the formation of larger ice crystals”? This phenomenon is due to the presence of water in system and not of MBA! Please make the adequate corrections!
3.2. Aerogel infrared spectroscopy (FTIR)
- L. 218: “3.2. The Fourier-transform infrared spectroscopy (FTIR) of aerogels” and not “3.2. Aerogel infrared spectroscopy (FTIR)”!
- L. 233: Please use “S6C0” and not “NPA” in Figure 3!
- Please improve section 3.2 with specific FTIR data regarding to both natural polymers, cellulose and starch! The observed information does not have to be narrated, but they must be demonstrated by the presence of the characteristic FTIR absorption bands. Please discuss all the FTIR bands present in the spectra and highlight the formation of chemical cross-linking bonds between the components! In addition, the authors must also discuss the bands in the wavenumber region 1500-500 cm-1!
3.3. Textural analysis of aerogels
- L. 237: “mass spectrometer (TPA)”?? There is no information related to TPA!
- There are no experimental data related to “hardness, elasticity, and cohesion” and no measurement units!
3.4. Dissolution of composite aerogels at different temperatures properties
- Dissolution is not the same phenomenon as swelling! In the title of section 3.4 "dissolution" appears, but the authors discuss the swelling of evidence! Please revise the section title!
- In Figure 7 and 8 the data are expressed in “mg/g” and “g/g” but the authors do not mention what these units of measure refer to! Please add all the information!
In my opinion, the manuscript still needs improvements and must be rigorously checked before to be recommend for publication in Polymers journal! There are serious difficulties in the interpretation of data and some of the foundations are unsafe.
Comments on the Quality of English LanguageModerate editing of English language required.
Reviewer 3 Report
Comments and Suggestions for Authors
The work is dedicated to production of starch-cellulose composite aerogels. Such composites can be used as adsorbents and drug carriers. The advantage of the material proposed is its composition: two natural polysaccharides. But this work still needs to be improved. First of all, the introduction should be expanded and the English should be improved. Here are some more comments:
1. The experimental section is missing the apparatuses. The section should be rewritten with the indication of the devices, the experimental conditions.
2. The letters on the SEM images are not visible.
3. The Results and Discussion section is written well, however all figures should be “changed”, since it is difficult to understand the information presented.
The manuscript is interesting, however it should be improved.
Comments on the Quality of English LanguageIn my point of view, the english editing is required.
Round 2
Reviewer 1 Report
Comments and Suggestions for Authors
The manuscript is acceptable.
Reviewer 2 Report
Comments and Suggestions for Authors
In my opinion, even if the authors paid attention to some of the criticized parts, the manuscript still have issues which must be corrected:
- L. 2: “property determination”! In my opinion, it is enough to write “properties”! The determination of them it stands to reason! Thus, “The Preparation and Properties of ...” or “Preparation and characterization” can be an option!
- L. 74-76: “it is necessary to use substances that can disrupt the crystalline arrangement of cellulose, which were often toxic, highly polluting, or expensive [22]. However, many of these substances are toxic, highly polluting, or come at a significant cost”! Please remove the repetition!
- L. 69-79: The part about aerogels prepared from cellulose is poorly presented, without any reference in the field, without highlighting their major importance in different fields of application! Furthermore, the advantages of this type of aerogels and their disadvantages are not mentioned, as is done for starch!
- L. 92: “Cellulose powder (65μm)”? Please mention “particle size of …”!
- L. 139-149: The section “2.2.5. Swelling properties of composite aerogels” is ambiguous presented! The authors make a confusion between "swelling power" and "swelling ratio", notions that are established by different methods! Please clarify the determination method and specify what you determine!
- In addition, please write clearly and separately all the equations used in this manuscript, a requirement that was also requested by the reviewer!
- Please explain the characterization method in scientific terms! What does "The expansion of the composite gel" mean? Are the authors referring to the increase of the volume of the sample?
- L. 147-148: "This process was repeated to measure the expansion of the composite gel at 70 and 90°C"? What are the four temperatures at which the measurements were made? It is not clear from the statement presented by the authors (25, 70 and 90°C)!
- L. 167-181: As I said before, the authors did not determine "the loading capacity of the aerogels"! First of all, they do not mention the time used for immersing the aerogel in the vitamin solution! Secondly, they do not mention whether the samples were dried after incorporation. Moreover, they can speak of inclusion capacity (as tested by them), only if it is considered that the diffusion of vitamin E in the aerogel is complete, so that the concentration of the vitamin in ethanol (in the impregnation bath) is the same as that in the aerogel, which is difficult to believe.
- L. 278: How did the authors determine Hardness in “g”, when this is measured in “N”?
- L. 255-306: The authors must discuss textural properties of the samples presenting clear data in section 3.3! There is no date mentioned in the text!
- L. 306: “different doses of crosslinker”? What do the authors mean by "dose"? Please use the correct terminology!
- L. 308: “at different temperatures properties”? “Temperatures” and not “temperatures properties”!
- L. 308-339: What do the authors mean by "Swelling power"? This is a different characteristic compared to the swelling degree or swelling ratio, which are determined by specific measurements! Please review carefully the entire section 3.4!
Comments on the Quality of English LanguageModerate editing of English language required.
Round 3
Reviewer 2 Report
Comments and Suggestions for Authors
Even if the authors have extensively explained in the “Responses to the comments” all the changes made to the Introduction section, described even on points (from 1 to 3), in fact, the authors have not changed a single word compared to the first version of this manuscript!!! They only changed the color of the text to red and nothing more!
The authors do not know the basic notions and make confusion between notions such as: swelling degree, swelling ratio and swelling power! What the authors determined, according to the method presented by them (2.2.5. Swelling properties of composite aerogels) is not the degree of swelling, but the swelling ratio!
The difference between swelling ratio and swelling degree is:
The swelling ratio (g/g) is determined by dividing the weight of the swollen hydrogel by the weight of the initial dried sample.
The swelling degree (%) = [(Ws – Wd)/Wd] × 100, where: Ws - the weight of the swollen hydrogels, at certain time; Wd - the initial dry weight of the sample.
Moreover, even if I clearly explained to the authors that they are confusing the process of vitamin E releasing with that of incorporating the active substance and that the authors determined the percentage of the vitamin E released from the matrix and not the amount of vitamin E incorporated into the matrix, the authors did not make any corrections. What's more, they even left the text intact! As in the case of the Introduction section, the only change made in the section "3.4. Swelling properties of composite aerogels at different temperatures" is the change of the color of the text from black to red!
In conclusion, I would like to emphasize one important thing, namely that the time of the reviewers is as important as that of the authors and that the superficial treatment by the authors of the comments made by the reviewers is nothing but a lack of respect towards the reviewers!
Comments on the Quality of English LanguageModerate editing of English language required.